# INFORM : Information eNtropy based multi-step reasoning FOR large language Models

**Chuyue Zhou**[1][*] **Wangjie You**[1][*] **Juntao Li**[1][†] **Jing Ye**[1]**, Kehai Chen**[2]**, Min Zhang**[1]

[1]Institute of Computer Science and Technology, Soochow University, China
[2] Harbin Institute of Technology, Shenzhen
{cyzhou, wjyouu, jyeyj}@stu.suda.edu.cn
{ljt, minzhang}@suda.edu.cn
chenkehai@hit.edu.cn

## Abstract

Large language models (LLMs) have demonstrated exceptional performance in reasoning tasks with dedicated Chain-of-Thought (CoT) prompts. Further enhancing CoT prompts with exquisite exemplars can significantly improve reasoning performance. However, the effectiveness of CoT prompts may fluctuate dramatically with different choices of in-context examples. Additionally, manual construction of rationale steps can be time-consuming, presenting challenges for the widespread adoption of CoT prompting. In this work, we propose a novel approach by introducing information entropy (IE) as a criteria on for CoT prompt selection. We extend this criterion to the CoT generation and inference stages, automatically generating CoT prompts with higher information entropy scores and adaptively determining the number of samples. These three stages together form our proposed information entropy based multi-step reasoning for large language models, named INFORM. Our experiments across seven reasoning benchmarks utilizing two language models(GPT-3.5-Turbo and text-davinci-003) demonstrate the superiority of INFORM both in performance and efficiency.[1]

## 1  Introduction

Large language models (LLMs) (Brown et al., 2020; Chowdhery et al., 2022; Thoppilan et al., 2022; Le Scao et al., 2022; Touvron et al., 2023) have achieved great success in recent years. These models are commonly employed through in-context learning (Brown et al., 2020), where instructions and exemplars are provided to enhance language understanding and generation. However, the widely-used in-context learning methods might perform poorly for complex reasoning tasks (Liang et al., 2022; Wei et al., 2022). Recent studies (Wei et al., 2022; Wang et al., 2023) have highlighted the importance of elaborating the reasoning steps in exemplars, leading to the emergence of chain-of-thought (CoT) prompting. CoT has shown promising results in improving the reasoning abilities of LLMs. Various strategies like self-notes (Lanchantin et al., 2023), progressive-hint prompting (Narang et al., 2023) and Least-to-Most prompting (Chowdhery et al., 2023), have been proposed to enhance CoT further.

While these works have focused on leveraging the inherent capabilities of models to augment the original prompts, recent research has shown that the performance of CoT heavily relies on the choice of exemplars (Lu et al., 2022; Liu et al., 2022; Wei et al., 2022; Zhang et al., 2023; Fu et al., 2023). This has led to investigations into identifying exemplars that maximize LLMs' reasoning abilities. Notably, complexity-based prompting (Fu et al., 2023) has introduced a selection strategy based on the number of rationale steps in annotated data. They have observed that more complex CoT prompts effectively stimulate LLMs' multi-step reasoning capabilities. However, this approach is limited by its dependency on carefully annotated data. In the absence of annotations, it degrades to a querylen-based strategy, which is more fragile to input noise.

This limitation highlights a fundamental drawback of CoT prompting, i.e., the heavy reliance on human engineering and the scarcity of annotated datasets, which are time-consuming to create. To address this, previous works such as zero-shot-cot (Kojima et al., 2022) and Auto-CoT (Zhang et al., 2023) attempted to alleviate the reliance on human effort in constructing CoT prompts. However, they faced challenges such as low-quality generation and high computational cost.

To overcome the aforementioned issues, we proposed a comprehensive CoT prompting process that bypasses the need for extensive human effort while improving the effectiveness of CoT. Our first

---

[*] Equal Contribution
[†] Juntao Li is the corresponding author.
[1]https://github.com/oneningt/INFORM

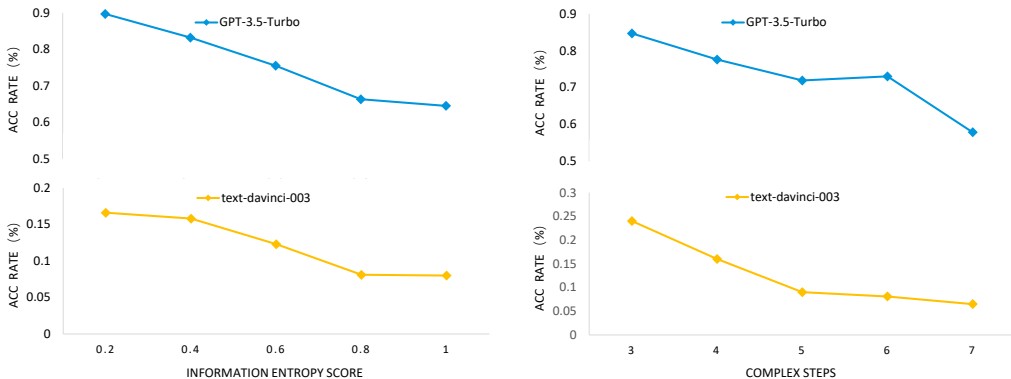

Figure 1: We compared the performance (ACC rate) of GSM8K dataset along with the increase of complex reasoning step and the information entropy scores on two models: GPT-3.5-Turbo and text-davinci-003.

and foremost objective is to identify a pre-selection strategy that achieves comparable effectiveness to complexity-based prompting while remaining applicable in all scenarios. Through extensive experiments, we have discovered that information entropy can serve as an effective and accurate criterion. Figure 1 depicts the similar trends between performance and the number of rationale steps and information entropy scores of queries, respectively, on the GSM8K dataset using GPT-3.5-Turbo and text-davinci-003. We consider information entropy to be a suitable criterion for several reasons: (1) Information entropy can be applied to queries, making it applicable to all datasets. (2) The information entropy score of a query provides insights into its complexity, which is related to the information contained in the annotated or generated prompt. (3) Information entropy is well-suited for both the generation and inference stages, enabling a coherent and unified framework, namely INFORM. Our experiments further validate the effectiveness of information entropy. Besides, we combined our information entropy criteria with self-consistency (Wang et al., 2023), extending our strategy from the input space (question selection) to the output space (CoT generation and IE inference).

In a nutshell, the contributions of our work are:

- We introduce information entropy as a new criterion for CoT prompts selection, improving LLMs' performance on reasoning tasks.

- We further apply information entropy criteria to the inference stage, generate reliable outputs automatically and save computation costs in the meantime

- We propose INFORM, a comprehensive and

effective CoT prompting framework consisting of three stages: question selection, CoT generation and information entropy self-consistency inference.

- Experimental results demonstrate the effectiveness of INFORM on different reasoning tasks utilizing different models.

## 2 Related Work

### 2.1 In-context Learning

Recent studies have shown that with the ever-increasing scale of language models, they show a remarkable ability to perform in-context learning (ICL) on downstream tasks (Brown et al., 2020; Kaplan et al., 2020). ICL enables LLMs to tackle a target task by using a few prompted examples as part of the input, allowing them to solve universal tasks without the need for gradient updates. However, it has been demonstrated that the performance of ICL is influenced by the prompts used (Liu et al., 2022, 2023). Therefore, determining the optimal prompt is a crucial and fundamental area of research.

### 2.2 Chain-of-Thought Prompt

Chain-of-Thought(CoT) is a novel prompting paradigm proposed by Wei et al. (2022), which involves a series of rationale steps leading to the final answer. CoT has been shown to significantly enhance performance on complex reasoning tasks such as arithmetic and commonsense reasoning. This success has spurred several subsequent works that adapt different strategies to improve CoT, including self-consistency (Wang et al., 2023), Least-to-Most prompting (Chowdhery et al., 2023),

self-notes (Lanchantin et al., 2023), Progressive-Hint Prompting (Narang et al., 2023), and self-polish (Wei et al., 2023).

Despite the remarkable success of CoT prompting, previous studies have primarily focused on how to use CoT to achieve the best results while ignoring how to construct prompting examples. Research by Liu et al. (2023) and Lu et al. (2022) has shown that the effectiveness of CoT prompting can vary widely depending on the choice of CoT examples. Given that data with manually annotated reasoning chains are scarce, and it is time-consuming to annotate rational chains manually in addition to the selection of questions and the accuracy of annotations that need to be considered, constructing appropriate prompts has been identified as a critical aspect of CoT prompting.

## 2.3 Example Construction For Prompting

Previous works on example construction for prompting can be viewed from two perspectives: rule-based selection and automated generation.

**Rule-based Selection** Rule-based selection methods employ various criteria to select optimal prompts from the original space. One intuitive strategy is to choose the most similar examples to the test question as its prompts. Nie et al. (2022) use nearest neighbor calibration to adjust the similarity scores between the input and the few-shot examples. KATE (Liu et al., 2022) shared the same strategy but scored the similarity based on the distance among embeddings. However, **similarity-based** strategies are usually computationally expensive. As an alternative, fairness-guided prompting (Ma et al., 2023) employs a content-free strategy that uses fairness as a metric to evaluate the predictive bias of a fixed prompt and shows that model performance is highly consistent with fairness. For CoT prompting, **fairness-based** methods require reasoning chain annotations for the whole training set, which compromises their advantage of being few-shot. Fu et al. (2023) propose that the reasoning ability of LLMs can be elicited by more complex CoT prompts, and the number of CoT steps can determine the complexity of the prompt. Their experiments demonstrate that selecting more complex CoT examples helps LLMs solve complex reasoning tasks. However, their **complexity-based** criteria heavily rely on labeled CoT data and degenerate to **querylen-based** when there is no labeled CoT data. In addition, Diao et al. (Diao et al.,

2023) proposes an **uncertainty-based** active selection strategy to determine which questions are the most important and helpful to annotate from a pool of task-specific queries.

In the realm of rule-based selection, most current methods tend to focus on searching prompts along a single dimension and either excessively rely on labeled data or require manual prompt construction. Our work sits in CoT prompting and proposes a novel IE-based selection strategy that evaluates queries from multi-dimension and constructs the CoT prompt automatically.

**Automated Generation** It is observed that data with annotated reasoning chains are scarce, and it's time-consuming to annotate manually. Some researchers have committed to automating the generation of CoT prompts for language models.

Auto-CoT (Zhang et al., 2023) classify questions into different clusters, select representative questions with diversity, and generate reasoning chains by zero-shot CoT prompt to construct demonstrations. Huang et al. (2022) demonstrate that LLM is also capable of self-improving with only unlabeled datasets. They use a pre-trained LLM to generate high-confidence rationale-augmented answers for unlabeled questions using Chain-of-Thought prompting and self-consistency and fine-tune the LLM using those self-generated solutions as target outputs. Automate-CoT (Shum et al., 2023) first augment rational chains from a small labeled dataset, then prune low-quality chains to construct a candidate pool of machine-generated rationale chains based on the labels, and finally apply a variance-reduced policy gradient strategy to estimate the gradients and optimize the latent variables with estimated gradients to select CoT.

These methods have extremely dispensed human efforts in generating CoT examples; however, most of them suffer from extra training costs, dependency on annotated data, and ignorance of the selection of questions. Our work proposes a comprehensive process of CoT prompting, covering rule-based selection of query, automatical generation of rationale steps and improvement of inference results.

## 3 INFORM

The overall schematic of our proposed INFORM are illustrated in Figure 2. Our approach is developed based on Information Entropy criteria, which we integrate into three different stages of standard

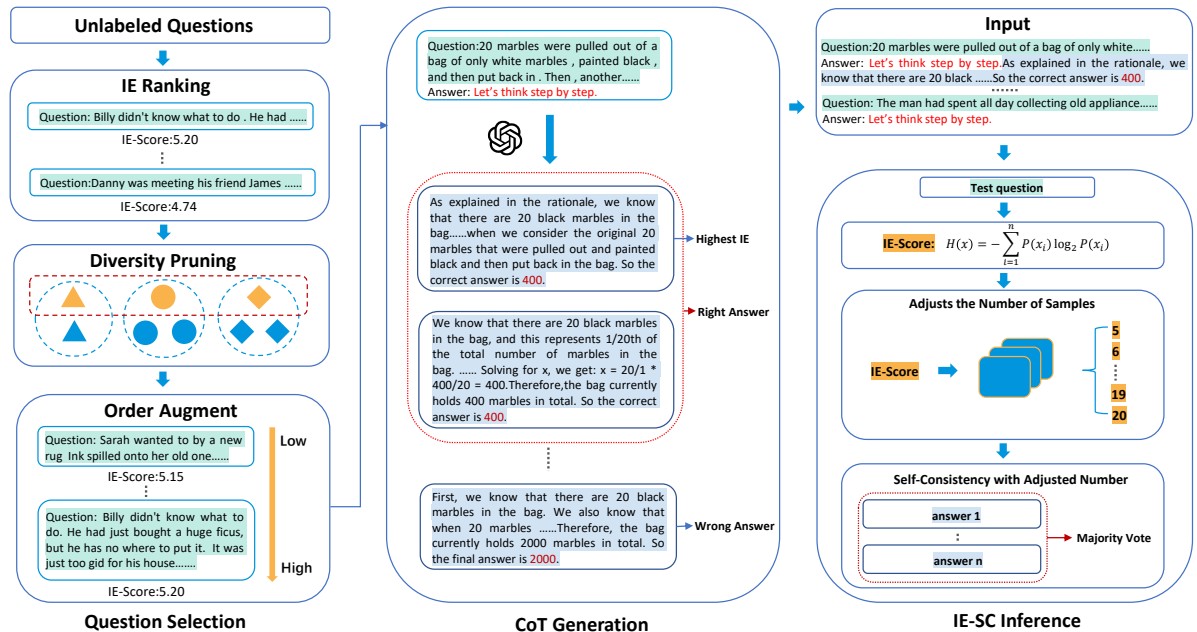

Figure 2: Overview of our proposed INFORM. **Question Selection**: Comprises three sub-steps: IE ranking, diversity pruning, and order augmentation, aimed at selecting informative questions. **CoT Generation**: Automatically generates rationale steps for the selected questions and identifies the one with the highest IE score. **IE-SC Inference**: Adjusts the number of samples based on the IE score of the query, followed by applying the original self-consistency method to conduct a majority vote.

few-shot CoT prompting. Specifically, our proposed approach consists of three sequential stages: question selection, CoT generation and IE-SC inference. We will introduce every stage in detail in the following sections.

## 3.1 Information Entropy

Information entropy describes the uncertainty or randomness of information. The definition of information entropy is as follows:

$$H(X) = -\sum_{i=1}^{n} p(x_i) \log_2 p(x_i) \quad (1)$$

The term $p(x_i)$ represents the probability of each outcome $x_i$. Therefore, as the information entropy increases, the sentence becomes more informative. As demonstrated in the complexity-based strategy, the more complex exemplars can significantly improve the performance of large language models.

## 3.2 Question Selection

Previous research consistently demonstrates that selecting exemplars based on specific criteria can significantly improve the performance of LLMs in multi-step reasoning tasks. We emphasize the importance of question selection instead of exemplar

selection. Specifically, our question selection strategy comprises three steps: IE ranking, diversity pruning and order augment.

**IE Ranking** Given a set of unlabeled questions $Q = (q_1, q_2, q_3...q_n)$, we calculate their information entropy score denoted by $H = (h_1, h_2, h_3...h_n)$, using Equation 1. We then select the top-$k$ questions with the highest IE scores, as we believe they will generate more informative rationale steps. In our experiments, we set $k$ to three times the number of candidates in the dataset.

**Diversity Pruning** After obtaining a set of questions with high IE scores, we consider incorporating a filtering step to enhance diversity. Specifically, we calculate the cosine-similarity among the question set with:

$$cos(\theta) = \frac{\sum_{i=1}^{n} (x_i \times y_i)}{\sum_{i=1}^{n} (x_i)^2 \times \sum_{i=1}^{n} (y_i)^2} \quad (2)$$

For examples with high similarity (>80%), we only keep the one with the highest IE score, discarding the rest.

**Order Augment** As widely recognized, in-context learning is sensitive to the order of demonstrations within prompts (Lu et al., 2022; Liu et al.,

2022). Intuitively, we sort the order of questions based on their information entropy score, effectively arranging them from easy to hard, aiming to exploit LLMs' rational reasoning ability gradually, which is similar to curriculum learning (Bengio et al., 2009) in spirit.

### 3.3 CoT Generation

After employing our query selection strategy to acquire a specific set of queries, we proceed to automatically generate the corresponding rationale steps using the zero-shot-CoT technique (Kojima et al., 2022). Note that we generate rationale steps only when the data lacks annotated CoT; otherwise, we utilize the original rationale steps. Taking inspiration from previous work that leveraged multi-sampling to enhance LLMs' performance during the inference stage (Wang et al., 2023), we extend this approach to our CoT generation stage. Specifically, given a selected query $q$, we append "Let's think step by step" to the end of the query and ask the model to answer it k times. We set $k = 10$ in our experiments. Once we obtain 10 distinct CoT sequences, we select the CoTs that have correct answers based on the ground truth provided in the dataset. If the correct answer is not given, we adhere to the original self-consistency approach to employ a majority voting mechanism. Finally, we select the CoT sequence with the highest information entropy score among the selected correct answers.

### 3.4 Information Entropy Self-Consistency Inference

Wang et al. (2023) proposed Self-Consistency, which improves performance for CoT reasoning by employing multiple diverse reasoning chains and aggregating their outputs using a simple majority voting technique. However, this approach comes with a trade-off of the increased computational cost that the model must be prompted multiple times for the same question. As LLMs continue to grow in size, the cost becomes increasingly unacceptable. Our empirical observations revealed that having more candidates does not always yield better performance, and using a fixed number of samples for every question in a dataset can be inefficient, particularly for straightforward questions. To address these issues, we introduce adaptive Information Entropy Self-consistency(IE-SC). Unlike the conventional approach, IE-SC dynamically adjusts the number of samples $n$ for each query instead of

using a fixed budget for the entire dataset. The adjustment is based on the information entropy score of the query and can be expressed as follows:

$$N_i = N_{\min} + \frac{N_{max} - N_{min}}{H_{max} - H_{min}} * (H(q_i) - H_{\min}) \quad (3)$$

where $H(q_i)$ represents the IE-score of $i$-th query. $H_{min}$ and $H_{max}$ denote minimum and maximum IE scores, which we set according to the dataset, typically to 3 and 8. $N_{min}$ and $N_{max}$ indicate the minimum and maximum number of samples, respectively, which we set to 5 and 20 in our experiment. Our experiment demonstrates that IE-SC, in contrast to the original SC approach, is well-suited for different datasets and different language models, and effectively reduces the computational costs without compromising output quality.

## 4 Experiments

### 4.1 Experimental Settings

**Datasets**  We evaluate our approach across seven benchmarks, covering three categories of reasoning tasks: (i) Arithmetic Reasoning: GSM8K (Cobbe et al., 2021), MultiArith (Roy and Roth, 2015), Addsub (Hosseini et al., 2014), AQuA (Ling et al., 2017) (ii) Commonsense Reasoning: CommonsenseQA (Talmor et al., 2019) and StrategyQA (Geva et al., 2021) (iii) Temporal Reasoning: Date-Understanding (Zhang et al., 2021). Among these datasets, GSM8K, AQuA, and StrategyQA provide their own rationale steps within their training splits. Hence, for these datasets, we seamlessly integrate our strategy into the query selection and inference stages, leveraging the original rationales. For the remaining datasets, we first select the queries and then automatically generate the Chains of Thought (CoT) prompts. By default, we employ the test split for evaluation. In cases where test sets are not provided, we assess performance on the validation set instead.

**Implementation**  We utilize the text-davinci-003 and GPT-3.5-Turbo version of the public ChatGPT model from the OpenAI API with 175 billion parameters (Brown et al., 2020; Ouyang et al., 2022) for most of our experiments. We selected these two LLMs because they are the latest and most robust models available. For most datasets, queries are selected from the official training split. Given that some datasets only have the test split, we select queries from the validation split or from the test

split itself, excluding them from subsequent experiments. Following Wei et al. (2022), we set the number of demonstrations k to 8, except for AQuA(4), StrategyQA (6), and CSQA (7). In datasets lacking annotated reasoning steps, we automatically generate CoTs and select the one with the highest IE score among the correct answers. Following Kojima et al. (2022), we add "Let's think step by step" before the reasoning chains for all prompting schemes to improve the performance. During the standard decoding process, we set the temperature to 0 and employ a greedy search algorithm to obtain results. Under the IE-SC setting, we set the temperature to 0.7 and determine the number of generations adaptively based on the IE score of the query, ranging from 5 to 20.

**Baselines** We compare our IE-CoT with three baselines: manual-CoT, complex-CoT, and Auto-CoT. To ensure fairness between our method with other baselines, we use the same number of CoT examples in all methods and datasets. We implement these methods on our own, following the original settings in the corresponding paper.

- Manual-CoT (Wei et al., 2022) is a widely used naive method for CoT prompting construction, which provides 4 to 8 manual-written exemplars consisting of a series of manual-written intermediate reasoning steps.

- Complex-CoT (Fu et al., 2023) utilizes the number of reasoning steps as criteria to select more complex exemplars in the annotated data.

- Auto-CoT (Zhang et al., 2023) is an automatic CoT exemplars construction method that clusters questions with diversity and generates CoT prompts by zero-shot-CoT.

### 4.2 Results

The main results are displayed in Table 1. Overall, our method consistently matches or exceeds the performance of baselines on most datasets, utilizing two different large language models. Our IE-CoT achieves an average improvement of 3.28% and 2.87% compared to manual-CoT with GPT-3.5-Turbo and text-davinci-003, respectively. Moreover, under the IE-SC settings, the improvement further increased to 7.38% and 6.51%. These results unequivocally demonstrate the effectiveness

of our proposed approach. In the subsequent sections, we categorize the datasets based on the presence of annotated rationale steps and provide a detailed discussion of our findings.

**Annotated Dataset** For datasets obtaining reasoning annotations themselves, such as GSM8K, AQuA, and StrategyQA, we simply apply our strategy to the question selection and inference stage, utilizing the original rationale steps from the training set. In comparison to complexity-based prompting, dedicated to selecting examples in the annotated dataset, our approach outperforms it in most datasets using two models. One exception is GSM8K dataset with text-davinci-003, where our method slightly lags behind the complexity-based strategy by 0.68%. Upon conducting a thorough analysis of the reasons, we have discovered a plausible explanation that the GSM8K dataset is meticulously annotated and the questions exhibit a significant diversity, allowing the complexity-based selection strategy to achieve surprising results. Furthermore, under the IE-SC setting, our method demonstrates additional improvements of 7.12%, 9.37% and 4.17%, respectively, with GPT-3.5-Turbo. Compared to methods automatically generating the CoT prompt, like Auto-CoT, our approach effectively leverages the labeled data, eliminating the need for augmenting extra CoT exemplars. In Appendix A, we supplement the experimental evidence that using original rationale steps leads to better results compared to generating it via LLMs, while also saving costs. Meanwhile, our strategy outperforms Auto-CoT by 3.03% and 2.15% on average with these three datasets with GPT-3.5-Turbo and text-davinci-003, respectively. This discrepancy can be attributed to our strategy's selection of queries based on predefined rules and the original manually constructed rationale steps, which is more reliable compared to the zero-shot CoT generated in Auto-CoT.

**Unlabeled Dataset** Our method consistently showcases improvements across the remaining unlabeled datasets, namely MultiArith, Addsub, CSQA, and Date-Understanding. In contrast with the competitive baseline, Auto-CoT, which automatically generates CoT prompt using zero-shot-CoT, the same as our approach, our approach surpasses it on all datasets utilizing GPT-3.5-Turbo and text-davinci-003. The most notable advancements are observed in the Addsub and Date-

|  | GSM8K | MultiArith | Addsub | AQuA | CSQA | StrategyQA | DATE |
|---|---|---|---|---|---|---|---|
| **LaMDA-137B** | | | | | | | |
| Manual | 14.30 | 44.90 | 51.90 | 20.60 | 57.90 | 65.40 | 26.80 |
| Self-consistency | 27.70 | 75.70 | 63.50 | 26.80 | 63.10 | 67.80 | - |
| **PaLM-540B** | | | | | | | |
| Manual | 56.90 | 94.70 | 91.90 | 35.80 | 79.90 | 77.80 | 65.30 |
| Self-consistency | 74.40 | 99.30 | 93.70 | 48.30 | 80.70 | 81.60 | - |
| **GPT-3.5-Turbo** | | | | | | | |
| Manual | 79.59 | 97.33 | 89.11 | 55.51 | 71.08 | 60.21 | 77.60 |
| Complex | 80.89 | 97.66 | 92.65 | 56.29 | 73.46 | 64.73 | 75.20 |
| Auto-CoT | 76.62 | 96.28 | 88.22 | 54.53 | 72.84 | 62.28 | 79.50 |
| **IE-CoT(ours)** | 81.65 | 98.66 | 92.91 | 58.59 | 74.69 | 64.86 | 82.00 |
| + IE-SC | **88.77** | **99.16** | **94.68** | **67.96** | **76.50** | **69.03** | **86.00** |
| **text-davinci-003** | | | | | | | |
| Manual | 56.55 | 94.16 | 85.75 | 43.70 | 77.55 | 68.16 | 78.40 |
| Complex | 60.42 | 95.16 | 88.35 | 44.88 | 77.64 | 70.08 | 75.60 |
| Auto-CoT | 57.63 | 95.08 | 87.94 | 42.60 | 76.80 | 70.68 | 78.00 |
| **IE-CoT(ours)** | 59.74 | 96.00 | 89.10 | 45.66 | 78.10 | 70.95 | 84.80 |
| + IE-SC | **65.20** | **97.33** | **91.13** | **51.90** | **80.09** | **74.58** | **89.60** |

Table 1: The overall performance of INFORM and the comparison against existing methods under different models on seven reasoning tasks. Manual-CoT, Complex and Auto-CoT denote chain-of-thought (Wei et al., 2022), complexity-based prompting (Fu et al., 2023) and Auto-CoT (Zhang et al., 2023), respectively. IE-SC denotes our performance under the IE-SC setting. **Bold** denotes the best performance in performed methods. Underline denotes the second performance. LaMDA (Thoppilan et al., 2022) and PaLM (Chowdhery et al., 2022) are not accessible to the public, so their numbers are from corresponding papers. The remaining methods are our own implementation.

understanding datasets, where our proposed strategy outperforms Auto-CoT by 3.85% and 3.98% on average with GPT-3.5-Turbo and text-davinci-003. This can be attributed to the relatively simplified nature of queries in these two datasets. While automatically generated CoT prompts may provide limited information, our strategy diligently selects the most informative queries and prompts during the selection and generation stages, resulting in a more effective prompting effect than Auto-CoT. Noting that complexity-based strategy degrades to querylen-based when handling unlabeled datasets, leading to the selection of highly similar questions. It lagged behind 2.32% and 2.82% on average by our approach with GPT-3.5-Turbo and text-davinci-003, respectively. This serves as further evidence of the effectiveness and versatility of our approach.

## 5 Analysis

We conducted additional experiments to assess the effectiveness of INFORM and analyze the contributions of each component of INFORM. Due to the high cost associated with using the text-davinci-003 API, we primarily utilized GPT-3.5-Turbo for the additional experiments.

**Effects of Question Selection** As shown in Fig 2, the query selection stage involves three steps: IE

| Datasets | GSM8K | AQuA |
|---|---|---|
| Random | 79.59 | 55.51 |
| IE-low | 78.22 | 53.93 |
| IE-high | 80.42 | 56.69 |
| IE-high + div | 81.50 | 58.26 |
| IE-high + div + order | **81.65** | **58.59** |

Table 2: Effect of every step in the query selection stage.

|  | MultiA | Addsub | CSQA |
|---|---|---|---|
| Random | 98.33 | 92.31 | 72.76 |
| IE-low | 97.83 | 91.17 | 73.49 |
| IE-high | **98.66** | **92.91** | **74.69** |

Table 3: Effect of information entropy score in CoT generation stage.

ranking, diversity pruning, and order augment. To evaluate the effectiveness of each step, we conducted an ablation study by analyzing their individual contributions. As presented in Table 2, our method consistently achieved improvements on two datasets. Specifically, IE-low and IE-high refer to the selection of queries with low information entropy scores and high information entropy scores, respectively. A substantial average gap of 2.48% is observed between these two extreme contrasts. Additionally, IE-low lags behind randomly selected exemplars by an average of 1.47%, high-

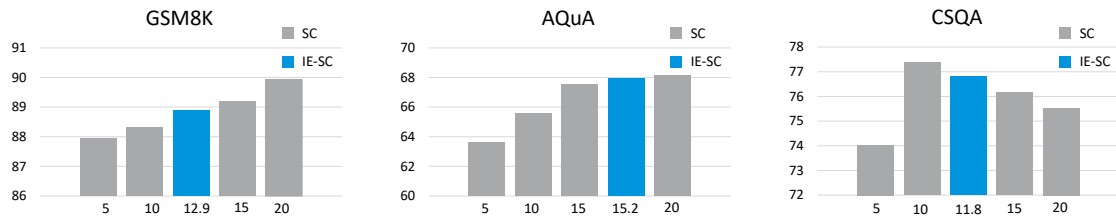

Figure 3: The performance of self-consistency with original self-consistency and our proposed IE-SC. The X-axis means the average number of candidates and the Y-axis means the exact accuracy.

| Models | BLOOM-7b | OPT-6.7b | Alpaca-7b | Alpaca-13b | Davinci-002 |
|---|---|---|---|---|---|
| Manual-CoT | 3.77 | 4.70 | 8.64 | 11.06 | 48.06 |
| IE-CoT | 4.25 | 5.48 | 9.70 | 17.21 | 50.49 |

Table 4: The performance of INFORM on five language models on GSM8K dataset.

lighting that queries with low information entropy lead to prompts lack of information, hampering the reasoning ability of LLMs. Moreover, the application of diversity pruning brings an additional 1.08% improvement for IE-high queries. This improvement can be attributed to the decreased similarity between queries, which provides diverse ideology to LLMs, enabling them to handle different types of challenging tasks. As widely recognized, in-context learning is sensitive to the order of demonstrations within prompts (Lu et al., 2022; Liu et al., 2022). Indeed, the result reveals that order-augment brings improvement; however, the magnitude of this improvement is very slight(0.24%). We leave further study on how to determine the optimal ordering of exemplars within prompts to fully exploit the capabilities of LLMs in future work.

**Effects of IE in CoT generation** After applying our query selection strategy to obtain a specific set of queries, we proceed to automatically generate the corresponding rationale steps using the zero-shot-CoT technique (Kojima et al., 2022). Noting that we generate rationale steps only when the data lacks annotated CoT. Therefore, we select three datasets, namely MultiA, Addsub, and CSQA, which do not have labeled rationales, to evaluate the effectiveness of choosing CoT prompts with high information entropy scores. As depicted in Table 3, selecting high information entropy (IE) score prompts consistently outperforms the other two strategies: random selection and selecting prompts with low IE scores. This observation suggests that CoT prompts with high IE scores are more effective in stimulating LLMs' reasoning ability, leading to improved performance.

**Effects of IE-SC** We conducted further analysis of our proposed Information Entropy Self-Consistency (IE-SC). As depicted in Fig 3, while self-consistency undeniably yields significant improvements for reasoning tasks, it is noteworthy that better performance does not necessarily rely on a larger candidate pool, as evidenced by the results on the CSQA dataset. Our proposed IE-SC approach achieves a remarkable 33% reduction in computational cost (equivalent to reducing 6.7 instances per query) compared to the original SC method, which samples 20 candidates. This reduction in cost comes with a minimal average decrease in accuracy of less than 0.2%. These findings demonstrate that our proposed approach strikes a favorable balance between effectiveness and efficiency.

**Robustness of INFORM** We performed additional experiments on various large language models to assess the robustness of our proposed INFORM approach. As illustrated in Table 4, our method consistently achieves improvements across different models, including the previous version of GPT-3.5 (text-davinci-002) as well as smaller-scale language models such as Alpaca-7b and Alpaca-13b. Since some larger-scale models are unavailable, i.e., Palm and LaMDA , we have not conducted experiments on them. Even so, these results have clearly demonstrated the versatility and effectiveness of INFORM, indicating its applicability to a wide range of language models. We further discuss the impact of linguistic characteristics on our approach and and provide some examples for further clarification in the appendix B and C.

## Acknowledgements

This work is supported by the National Science Foundation of China (NSFC No. 62276077 and 62206194), Shenzhen College Stability Support Plan under Grants GXWD20220811170358002 and GXWD20220817123150002, and the Natural Science Foundation of Jiangsu Province (Grant No. BK20220488).

## 6 Conclusion

This paper presents INFORM, a novel framework for CoT prompting that consists of a comprehensive process encompassing query selection, CoT generation, and IE-SC inference. INFORM is designed to be adaptable to various datasets and models, effectively improving the performance of CoT prompts and mitigating the requirement for additional human involvement. Our experimental results demonstrate the efficacy and versatility of INFORM, showcasing its ability to significantly enhance CoT performance in various scenarios.

## Limitations

Despite the promising results of our proposed INFORM, it comes with several limitations that should be addressed in future work:

- While queries are typically manually constructed, our query selection strategy may be susceptible to noise when the queries contain irrelevant information. Integrating additional criteria could help mitigate this issue.

- Our experiments focused solely on reasoning tasks, showcasing the effectiveness of INFORM with CoT prompts. However, the performance of INFORM on non-reasoning tasks remains unknown.

- Although we have evaluated the robustness of INFORM on several different LLMs of varying sizes, there may still be models, such as very large-scale models like Palm, where INFORM may not be as effective.

These limitations highlight areas for further investigation and refinement in order to enhance the applicability and performance of INFORM in a wider range of scenarios.

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

## A  Compared With Original CoT

To elucidate the implications of disregarding the included CoT strings and relying on INFORM instead, we conducted supplementary experiments using GPT-3.5-Turbo. The experimental setup involved three conditions: "INFORM-Without-CoT," which involved providing only the selected queries and their corresponding direct answers as examples without including the rationale steps; "INFORM-Generated-CoT," which entailed generating CoTs using INFORM for the selected questions, specifically through the CoT-generation operation; and "INFORM-Original-CoT," which employed the original CoTs included in the datasets as per our original strategy. The obtained results are presented below:

|  | GSM8K | AQuA | StrategyQA |
|---|---|---|---|
| Without-CoT | 63.47 | 55.69 | 52.14 |
| Generated-CoT | 75.18 | 56.71 | 62.58 |
| Orginal-CoT | 81.65 | 58.59 | 64.86 |

Table 5: Compare Generated-CoT with Orginal-CoT.

Based on the obtained results, it can be inferred that the Original-CoT strategy exhibited the most favorable outcome, indicating that a manually constructed CoT outperforms a zero-shot-CoT generated by the model in the majority of scenarios. Nevertheless, when compared to examples without CoTs, the automatically generated CoT still demonstrated a significant improvement.

## B  Linguistic Characteristics

It is important to notice that language/discourse-based reasoning, different from reasoning in general, may be affected by the linguistic characteristics of the language used. Our work has not explicitly addressed the syntactic, semantic, and pragmatic aspects of the language. The focus of our work, however, is on how to select or construct better CoT examples through simple analysis of the data in the training set rather than how linguistic characteristics would affect the performance of large models on reasoning tasks. Our findings can serve as a starting point for researchers interested in exploring the impact of language-specific features on reasoning tasks.

In addition, shi et al( 2022) have demonstrated that reasoning in English (EN-CoT) consistently achieves competitive or better performance than reasoning in the native language of the question, which implies that our method can generalize to other languages. We have conducted some additional experiments in other languages; we built three Chinese CoT datasets that were translated from English. The results are as follows:

|  | MultiA_zh | Strategyqa_cn | Addsub_cn |
|---|---|---|---|
| Manual-CoT | 96.43 | 68.14 | 81.20 |
| IE-CoT | 98.55 | 68.85 | 90.35 |

Table 6: Performance of IE-CoT in Chinese-CoT datasets.

## C  Case Study

When compared to alternative methodologies, such as the selection of queries based on complexity, the utilization of query-len in datasets lacking rationales may lead to the occurrence of invalid character placeholders, exemplified by the inclusion of

"Question: Shipment - - - No . of Defective Chips / shipment - - - Total Chips in shipment
S 1- - - - - - - - 3- - 8,000
S 2- - - - - - - 5- - 12,000
S 3- - - - - - - 6- - 18,000
S 4- - - - - - - 4- - 16,000
A computer chip manufacturer expects the ratio of the number of defective chips to the total number of chips in all future shipments to equal the corresponding ratio for shipments S 1 , S 2 , S 3 , and S 4 combined , as shown in the table above . What ' s the expected number of defective chips in a shipment of 60,000 chips ?
Options: (A) 14 (B) 22 (C) 20 (D) 24 (E) 25
A: Let's think step by step. for a total of 51000 chips ( adding S 1 , S 2 , S 3 , S 4) total number of defective chips is 17 ( adding defective chips of S 1 , S 2 , S 3 , S 4) so ratio is 18 / 54000 or 1 every 3000 chips . Keeping this ratio constant for 60000 chips number of defective chips will be ( 1 / 3000) * 60000 = 20 The answer is (C), 20."

For complex detection methods, special characters are used to define the steps of rationale, it is sensitive to different characters for different datasets.