# OpenReview forum: "INFORM : Information eNtropy based multi-step reasoning FOR large language Models"
_EMNLP/2023/Conference — EMNLP 2023 Main_

### Official Review · Reviewer_ZjNo · 2023-08-02

**Soundness:** 4

**Excitement:**

3: Ambivalent: It has merits (e.g., it reports state-of-the-art results, the idea is nice), but there are key weaknesses (e.g., it describes incremental work), and it can significantly benefit from another round of revision. However, I won't object to accepting it if my co-reviewers champion it.

**Paper Topic And Main Contributions:**

This paper develops a new method for selecting demonstrations and associated chain-of-thought elements. The method is based on the unigram-based entropy of these elements – more surprising elements are favored to be chosen, presumably because they have higher information content. The entropy of the test item is also used to sort the demonstations and to dynamically select the number of demonstrations, and CoT elements are filtered to those that get the correct answer. The results consistently strong along a number of QA datasets.

**Questions For The Authors:**

1. Can you confirm that the entropy calculations are done on the unigrams? It's not really clear from the paper, but this is what I inferred from the code. Why not use the model logits for text-davinci-003 at least?

2. How are the Manual results obtained for datasets without CoT examples included?

3. The paper is written as though CoT was central, but actually four of seven the datasets have CoT strings included, and so all that is happening is that demonstrations are being selected and ranked. What happens if the include CoT strings are not used?

4. Are the results all based on single runs? If so, what is the observed variation across runs and across choices for the many hyperparameters that are in play here? If these hyperparameters were tuned informally based on pilot experiments, what examples were used for that and how extensively was this done across all the approaches?

**Reasons To Accept:**

This is a very lightweight addition to the toolkit for selecting demonstations and CoT strings. If I understand correctly, the entropy calculations are done entirely on the whitespace tokenized versions of the strings, so it doesn't even require access to model logits.

The reported results are very strong, and the paper does a good job of seeking to assign credit to the various components of the system.

**Reasons To Reject:**

I do not have reasons to reject per se, but I do have come crucial question for the authors, and the answers might lead me to adjust my overall perspective.

The paper's evaluation should rest entirely on the results. In terms of scientific ideas: the system is just a pipeline of threshold-based decisions using a simply notion of entropy. This is great if it truly works.

**Reproducibility:**

2: Would be hard pressed to reproduce the results. The contribution depends on data that are simply not available outside the author's institution or consortium; not enough details are provided.

**Reviewer Confidence:**

4: Quite sure. I tried to check the important points carefully. It's unlikely, though conceivable, that I missed something that should affect my ratings.

---

> ### Author Rebuttal · Authors · 2023-08-29
>
> **We thank the reviewer for taking the time to evaluate our work and provide valuable feedback. Here, we address the main concerns that were brought up.**
>
> - ***Can you confirm that the entropy calculations are done on the unigrams? It's not really clear from the paper, but this is what I inferred from the code. Why not use the model logits for text-davinci-003 at least?***
>
>   - Your understanding is correct. Actually, we have tried several methods to compute information entropy in our early experiments, including using model logits. However, we ultimately chose to use the unigram approach for the following reasons:
>
>     1. Our experiments primarily focused on reasoning datasets containing numerous Arabic numerals and mathematical symbols. Logits from models that were not specifically trained on such data may not accurately capture the desired information. On the other hand, using unigrams can better reflect the information since it is more directly based on the frequency distribution of individual words.
>
>     2. Model logits for short sentences may exhibit numerical irregularities. When the input sentence is short, the model's output logits may be very small, causing the values of the exponential function in the softmax operation to be extremely close to zero. In such cases, numerical overflow may occur during the computation of softmax. For instance, when we used the model logits of  GPT2 to calculate the information entropy for the MultiA dataset,  the information entropy score would be NAN, which means using model logits is not a universally applicable method.
>
>     3. It's worth noting that we extend our method to context generation and final result selection stages. Therefore, we opted for a more efficient approach to ensure feasibility within our overall framework. Using unigrams for entropy calculations could be a deliberate choice due to certain trade-offs.
>
>
>     Based on the aforementioned three points, we have chosen the unigram approach for computing information entropy. However, we acknowledge that model logits can yield comparable results in certain scenarios and there may be other potentially superior methods for calculating information entropy.
>
>
>  - ***How are the Manual results obtained for datasets without CoT examples included?***
>
>    - The manual results for datasets without CoT examples included were obtained by randomly selecting examples from the training set and manually composing chains of thought for them as we demonstrated in the Baseline part in section 4.1 Experiment settings. This process was followed based on the approach described by Wei et al. [1]. There is an example in the MultiA dataset:
>
>      > Question: Tom bought 12 boxes of chocolate candy and gave 7 to his little brother. If each box has 6 pieces inside it, how many pieces did Tom still have?
>      >
>      > Answer: Let's think step by step.
>      > Tom had 12 boxes of candy originally. Then he gave 7 to his little brother, so he had 12 - 7 = 5 boxes of candy left. Each box has 6 pieces inside it, so Tom still had 5 x 6 = 30 pieces.
>      >
>      > Therefore, the answer (arabic numerals) is 30.
>
>
>
> - ***The paper is written as though CoT was central, but actually four of seven the datasets have CoT strings included, and so all that is happening is that demonstrations are being selected and ranked. What happens if the include CoT strings are not used?***
>
>   - As we mentioned in section 3.3, when the data lacks annotated CoT,  we utilize our selection strategy to acquire a specific set of queries first. Following the approach outlined by Kojima et al. [2], rationale steps are generated through zero-shot CoT. We further integrated our method into the process of CoT generation. The effectiveness was shown in the ''Effects of IE in CoT generation" part in section 5.
>     As for the case of not using CoT, there will be a significant collapse in the results, which has been extensively discussed and experimented in [1]. We also conducted experiments on GPT-3.5-Turbo and the results are as follows：
>
>     |              | Addsub | AQUA  | Strategyqa |
>     | ------------ | ------ | ----- | ---------- |
>     | one-shot     | 84.28  | 50.19 | 35.96      |
>     | one-shot-cot | 88.59  | 53.81 | 44.58    |
>
>
>
> - ***Are the results all based on single runs? If so, what is the observed variation across runs and across choices for the many hyperparameters that are in play here? If these hyperparameters were tuned informally based on pilot experiments, what examples were used for that and how extensively was this done across all the approaches?***
>
>   - Yes, most of our experiments were conducted based on single runs. In the absence of self-consistency setting, we set the temperature of models to 0, and there was no additional randomness in our example selection stage, which means that for a specific dataset, our inputs remained the same for each run, resulting in deterministic outcomes. However, in the self-consistency setting, we still relied on single runs as we believe that the self-consistency setup itself partially eliminates randomness and also helps in computational efficiency.
>
>    - Regarding hyperparameters, we mostly followed the experimental settings of Wei et al.[1] and Wang et al.[3], such as the number of examples used for each dataset, temperature settings, and the minimum sampling count for self-consistency. These hyperparameters have been explored in their paper to understand their impact on the results.
>    - For the selection of H_min, H_max, N_min, and N_max. As mentioned after Equation 3 in our paper, H_min and H_max represent the minimum and maximum IE scores of the candidates, respectively. These values are calculated by equation 1 on different datasets. Typically, we have observed that the values are 3 and 8.
>
>       Regarding N_min and N_max, they indicate the minimum and maximum numbers of samplings for self-consistency. In our experiment, we followed the approach proposed by Wang et al. [3] and selected 5 as the value for N_min. The reason we selected 20 for N_max was based on our previous experiments, where we found that increasing the number of samplings beyond 20 did not yield significant further improvements in self-consistency. We believe that these values strike a balance between computational efficiency and achieving desirable results. We appreciate the reviewer's inquiry, and we hope this explanation clarifies the rationale behind our choices.
>    - Additionally, there are other hyperparameters such as the number of CoT generations and the threshold for similarity pruning, which were set empirically. However, the focus of this paper is not on tuning these hyperparameters in every stage for their subtle effects on the results, but rather on exploring the contributions of each component. Perhaps in future work, more in-depth research can be conducted on these hyperparameters.
>
> ***We hope we have addressed all the questions and thank you again for your valuable feedback. Please let us know if there are any additional concerns***
>
> [1] Wei J, Wang X, Schuurmans D, et al. Chain-of-thought prompting elicits reasoning in large language models[J]. Advances in Neural Information Processing Systems, 2022, 35: 24824-24837.
>
> [2] Kojima T, Gu S S, Reid M, et al. Large language models are zero-shot reasoners[J]. Advances in neural information processing systems, 2022, 35: 22199-22213.
>
> [3] Wang X, Wei J, Schuurmans D, et al. Self-Consistency Improves Chain of Thought Reasoning in Language Models[C]//The Eleventh International Conference on Learning Representations. 2023.

---

### Official Review · Reviewer_tD1V · 2023-08-04

**Typos Grammar Style And Presentation Improvements:** N.A.
**Soundness:** 4

**Excitement:**

4: Strong: This paper deepens the understanding of some phenomenon or lowers the barriers to an existing research direction.

**Missing References:**

N.A.

**Paper Topic And Main Contributions:**

Large Language Models (LLMs) have shown remarkable performance in reasoning tasks using Chain-of-Thought (CoT) prompts. The effectiveness of these prompts can be enhanced with specific examples but may vary widely, and manually constructing them is challenging. This work proposes a novel approach, named INFORM, which employs information entropy (IE) as a criterion for CoT prompt selection, extension, and inference. This method automatically generates CoT prompts with higher information entropy scores, and adaptively determines the number of samples, leading to a three-stage, information-entropy-based reasoning process. INFORM has been tested across seven reasoning benchmarks using two different models, GPT-3.5-Turbo and text-davinci-003, showing its superiority in both performance and efficiency. The key contributions include the introduction of information entropy to CoT prompts, its application to the inference stage, and the development of INFORM, a comprehensive framework for CoT prompting, which has proven effective in different reasoning tasks and models.

**Questions For The Authors:**

1. In the subsection Robustness of INFORM, which benchmark do you refer to?
2. How do you derive the values for H_min, H_max, N_min, and N_max?
3. Could you kindly specify the whole process of how you apply Equation 1 in the process of IE Ranking?

**Reasons To Accept:**

1. I like the idea of Question Selection: using information entropy ranking, diversity pruning, and order augmentation, aimed at selecting informative questions. In particular, I really like the idea of using information entropy in this case.
2. The authors conducted an ablation study to evaluate the effectiveness of each component of the proposed framework.
3. The proposed framework worked well across various reasoning tasks.

**Reasons To Reject:**

1. In the subsection entitled "Robustness of INFORM," the author neglects to specify the benchmark used in evaluating the performance of GPT-3.5 (text-davinci-002), Alpaca-7b, and Alpaca-13b.

2. The author has not provided details regarding the derivation of the values for H_min, H_max, N_min, and N_max.

3. It was not very clear how the authors derive p($x _i$) when applying Equation 1 in IE Ranking. How do you determine the probability of each outcome in this process?

**Reproducibility:**

4: Could mostly reproduce the results, but there may be some variation because of sample variance or minor variations in their interpretation of the protocol or method.

**Reviewer Confidence:**

5: Positive that my evaluation is correct. I read the paper very carefully and I am very familiar with related work.

---

> ### Author Rebuttal · Authors · 2023-08-29
>
> **We thank the reviewer for taking the time to evaluate our work and provide valuable feedback. Here, we address the main concerns that were brought up.**
>
> - ***In the subsection Robustness of INFORM, which benchmark do you refer to?***
>
>   - The evaluation was conducted on the GSM8K benchmark in the "Robustness of INFORM" section. We understand that this was not explicitly stated, and we apologize for any confusion caused.
>
> - ***How do you derive the values for H_min, H_max, N_min, and N_max?***
>
>   - As mentioned after Equation 3 in our paper, H_min and H_max represent the minimum and maximum IE scores of the candidates, respectively. These values are calculated by equation 1 on different datasets. Typically, we have observed that the values are 3 and 8.
>
>   - Regarding N_min and N_max, they indicate the minimum and maximum numbers of samplings for self-consistency. In our experiment, we followed the approach proposed by Wang et al. [1] and selected 5 as the value for N_min. And the reason we selected 20 for N_max was based on our previous experiments, where we found that increasing the number of samplings beyond 20 did not yield significant further improvements in self-consistency. We believe that these values strike a balance between computational efficiency and achieving desirable results. We appreciate the reviewer's inquiry, and we hope this explanation clarifies the rationale behind our choices.
>
>
>
> - ***Could you kindly specify the whole process of how you apply Equation 1 in the process of IE Ranking?***
>
>   - Equation 1 represents the calculation of the information entropy score.  We have experimented with several methods to calculate information entropy in our early experiments, and we ultimately chose the unigram method. Specifically, we preserved the original sentences without tokenizing or removing special characters since reasoning datasets often contain many Arabic numerals and mathematical symbols, which are important signals reflecting the amount of information. Then, we calculated the frequency of each word appearing in the sentence, denoted as p(xi), to obtain the final result of Equation 1. We consider this as the information entropy score of the sentence. We sorted all the queries in the training set based on their information entropy scores and selected the ones with higher scores, which we refer to as IE-ranking, indicating that they contain richer information.
>
> ***We hope we have addressed all the questions and thank you again for your valuable feedback. Please let us know if there are any additional concerns.***
>
>
> [1] Wang X, Wei J, Schuurmans D, et al. Self-Consistency Improves Chain of Thought Reasoning in Language Models[C]//The Eleventh International Conference on Learning Representations. 2023.

---

### Official Review · Reviewer_ZqC6 · 2023-08-05

**Soundness:** 4

**Excitement:**

4: Strong: This paper deepens the understanding of some phenomenon or lowers the barriers to an existing research direction.

**Paper Topic And Main Contributions:**

The present paper proposes a method, for example, selection and automatic CoT prompt-generation for in-context learning in large language models based on the measure of information entropy.
The authors compare the notion of information entropy with the notion of complexity previously proposed by  Fu et al., showing that prompting based on information entropy approximates the behaviour of prompting based on complexity. The authors use this reasoning to propose information entropy as a measure for prompt selection in the training stage.
Further, the authors extend their idea and employ information entropy for the generation of CoT prompts for datasets which do not contain the rationale for the answers, pulling from the work on Auto-CoT of (Zhang et al. In this setting, the authors employ information entropy as a measure of quality for the selection of CoT prompts generated. Further, they propose the use of entropy at inference time to autonomously adjust the number of samples extracted from the model to answer the question in a self-consistency approach.
The authors perform an evaluation of their approach on four non-open LLM over seven different reasoning tasks, comparing their approach with previous methods such as prompting by complexity, auto-cot and manual prompts.

They perform analyses on the impact of entropy on the query selection stage and CoT generation. They evaluate the method on two small open-source language models (namely alpaca 7b and alpaca 13b), but it is not clear on which task.

**Questions For The Authors:**

On what task has the evaluation on open-source models conducted, and why is gpt-3.5-Turbo listed as open-source?
On which open-source models do the authors evaluate their framework?

**Reasons To Accept:**

I find the idea of using Information Entropy for the selection of prompts interesting in its own right as it can help investigate the information-theoretic connections of in-context learning, and the results obtained showing the entropy helps the model learn in complex tasks points to an interesting question on the theory of such large language models.
Aside from the measure itself, the authors propose a promising prompting framework for multi-step reasoning that could be implemented based on different other measures, even example complexity itself.
The paper is very clear, and the empirical validation of the proposal is quite thorough, in my opinion.

**Reasons To Reject:**

The authors claim to evaluate their method on various open-source models but do not explain which models were evaluated, only the small alpaca models. It seems odd not to compare with available larger open-source language models such as  OPT and BLOOM.
The authors do not report on budget and CO2 emissions, which is an important aspect of research on LLMs

It is important to notice that language/discourse-based reasoning, different from reasoning in general, may be affected by the linguistic characteristics of the language used in the work. While the work solely relies on reasoning in English, the authors fail to acknowledge or consider how syntactic, semantic and pragmatic aspects of the language may have impacted their work and whether their results may generalise to other languages.

**Reproducibility:**

4: Could mostly reproduce the results, but there may be some variation because of sample variance or minor variations in their interpretation of the protocol or method.

**Reviewer Confidence:**

3: Pretty sure, but there's a chance I missed something. Although I have a good feel for this area in general, I did not carefully check the paper's details, e.g., the math, experimental design, or novelty.

**Typos Grammar Style And Presentation Improvements:**

pg 2 line 109: we extend apply -> we extend?
pg 2 line 116: Cot generation -> CoT generation
 pg 7 Table 2: Rondom -> Random

---

> ### Author Rebuttal · Authors · 2023-08-29
>
> ***We thank the reviewer for taking the time to evaluate our work and provide valuable feedback. Here, we address the main concerns that were brought up.***
>
> - ***The authors claim to evaluate their method on various open-source models but do not explain which models were evaluated, only the small alpaca models. It seems odd not to compare with available larger open-source language models such as OPT and BLOOM. The authors do not report on budget and CO2 emissions, which is an important aspect of research on LLMs***
>
>   - We appreciate the feedback regarding our evaluation of open-source models in our paper. We apologize for the confusion caused by our statement about evaluating on various open-source models, and a more accurate statement would be "evaluating on various models". While our main experiments were conducted on the 175B GPT-3.5 models, we believe it is valuable to further validate our approach on smaller-scale models, so we chose the small alpaca models. Regarding the OPT and BLOOM models, we conducted additional experiments on the GSM8K dataset and the results are as follows:
>
>     |            | BLOOM-7b | OPT-6.7b | OPT-13b |
>     | ---------- | -------- | -------- | ------- |
>     | Manual-CoT | 3.77     | 4.70     | 6.36    |
>     | IE-CoT     | 4.25     | 5.48     | 8.44    |
>
>    - We also understand the importance of considering budget and CO2 emissions in language model research. Unfortunately, **due to some policy restrictions**, we cannot provide a specific breakdown of our budget. However, we can estimate the cost based on the charging guidelines of the OpenAI API. Evaluating a 100-instance test set costs approximately 0.3-0.5 US dollars for greedy decoding (1 output chain). It's worth noting that the cost may vary depending on the number of tokens and the specific model used. In our analysis experiments, we primarily utilized the GPT-3.5-Turbo model, as it offers a more cost-effective solution compared to text-davinci-002. Regarding CO2 emissions, since our experiments primarily focus on the inference stage of LLMs and the GPT-3.5 models are closed-source, we lack specific information about the emissions. However, our IE-SC can effectively decrease the computational cost for self-consistency settings, as mentioned in our paper.
>
> - ***It is important to notice that language/discourse-based reasoning, different from reasoning in general, may be affected by the linguistic characteristics of the language used in the work. While the work solely relies on reasoning in English, the authors fail to acknowledge or consider how syntactic, semantic and pragmatic aspects of the language may have impacted their work and whether their results may generalise to other languages.***
>
>   - Thank you for bringing up these important points. It is indeed crucial the impact of linguistic characteristics on language/discourse-based reasoning. We acknowledge that our work has not explicitly addressed how syntactic, semantic, and pragmatic aspects of the language may have influenced our findings. The focus of this paper, however, is on how to select or construct better CoT examples through simple analysis of the data in the training set, rather than how linguistic characteristics would affect the performance of large models on reasoning tasks. Our findings can serve as a starting point for researchers interested in exploring the impact of language-specific features on reasoning tasks.
>
>   - In addition, shi et al.[1] have demonstrated that reasoning in English (EN-CoT) consistently achieves competitive or better performance than reasoning in the native language of the question, which implies that our method can generalize to other languages. We have conducted some additional experiments in other languages, we built three Chinese CoT datasets that were translated from English, the results are as follows:
>     |            | MultiA_cn | Strategyqa_cn | Addsub_cn |
>     | ---------- | --------- | ------------- | --------- |
>     | Manual-CoT | 96.43     | 68.14         | 81.20     |
>     | IE-CoT     | 98.55     | 68.85         | 90.35     |
>
>
> - ***Typos Grammar Style And Presentation Improvements:***
>
>    - We have corrected the other presentation/typos/grammatical errors listed in your comments. Thank you again for your patient and valuable proofreading.
>
>
> - ***On what task has the evaluation on open-source models conducted, and why is GPT-3.5-Turbo listed as open-source? On which open-source models do the authors evaluate their framework?***
>
>   - The evaluation was conducted on the GSM8K benchmark in the "Robustness of INFORM" section. Regarding the mention of GPT-3.5 models as open-source, we apologize for any confusion caused. Our intention was to highlight that we evaluated our framework on various models, including both open-source and close-source ones. We understand that this statement may have been misleading, and we apologize for the lack of clarity in our original paper, we will make the necessary corrections in the later version of the paper.
>
>     We have evaluated our framework on the alpaca-7b and alpaca-13b models. To further prove the robustness of INFORM, we have conducted additional experiments on OPT and BLOOM models and the results are as follows:
>
>     |     | BLOOM-7b | OPT-6.7b | OPT-13b | alpaca-7b | alpaca-13b |
>     | ---------- | -------- | -------- | ------- | --------- | ---------- |
>     | Manual-CoT | 3.77     | 4.70     | 6.36    | 8.64      | 11.06      |
>     | IE-CoT     | 4.25     | 5.48     | 8.44    | 9.70      | 17.21      |
>
>
>
> ***We hope we have addressed all the questions and thank you again for your valuable feedback. Please let us know if there are any additional concerns.***
>
> [1] Shi F, Suzgun M, Freitag M, et al. Language models are multilingual chain-of-thought reasoners[C]//The Eleventh International Conference on Learning Representations. 2023.
>
> ##

---

### Meta-Review · Area_Chair_ns1Y · 2023-09-16

**Recommendation:** 4

**Metareview:**

After spirited discussion, the reviewers agree that this is a sound paper with a surprising and interesting result: you can use entropy to select with chain-of-thought rationales are effective for QA tasks. Where there is less agreement among the reviewers is how excited we should be about this. While this is a useful tool, the method seems to be dependent on some important details not adequately specified in the submission (but addressed in the author response period) and thus hard to replicate, and this would be a stronger paper if there were more emphasis/discussion on why this works. However, many practitioners will be immensely interested in the "how", which may be enough.

---

### Decision · Program_Chairs · 2023-10-07

**Decision:**

Accept-Main

**Comment:**

After spirited discussion, the reviewers agree that this is a sound paper with a surprising and interesting result: you can use entropy to select with chain-of-thought rationales are effective for QA tasks. Where there is less agreement among the reviewers is how excited we should be about this. While this is a useful tool, the method seems to be dependent on some important details not adequately specified in the submission (but addressed in the author response period) and thus hard to replicate, and this would be a stronger paper if there were more emphasis/discussion on why this works. However, many practitioners will be immensely interested in the "how", which may be enough.